# Effects of Different Na^+^ Concentrations on cAMP-Dependent Protein Kinase Activity in Postmortem Meat

**DOI:** 10.3390/foods13111647

**Published:** 2024-05-24

**Authors:** Ying Xu, Xubo Song, Zhenyu Wang, Yuqiang Bai, Chi Ren, Chengli Hou, Xin Li, Dequan Zhang

**Affiliations:** Institute of Food Science and Technology, Chinese Academy of Agricultural Sciences/Key Laboratory of Agro-Products Quality & Safety in Harvest, Storage, Transportation, Management and Control, Ministry of Agriculture and Rural Affairs, Beijing 100193, China; xuying114886@163.com (Y.X.); s1789894263@163.com (X.S.); caasjgsmeat2021_1@126.com (Z.W.); yuqiangbai_1844@163.com (Y.B.); candymce@163.com (C.R.); houchengli@caas.cn (C.H.); dequan_zhang0118@126.com (D.Z.)

**Keywords:** meat, Na^+^, PKA activity, protein phosphorylation

## Abstract

cAMP-dependent protein kinase (PKA) activity regulates protein phosphorylation, with Na^+^ playing a crucial role in PKA activity. The aim of this study was to investigate the effects of different Na^+^ concentrations on PKA activity and protein phosphorylation level in postmortem muscle. The study consisted of two experiments: (1) NaCl of 0, 20, 100, 200 and 400 mM was added to a muscle homogenate incubation model to analyze the effect of Na^+^ concentration on PKA activity, and (2) the same concentrations were added to pure PKA in vitro incubation models at 4 °C to verify the effect of Na^+^ on PKA activity. The PKA activity of the muscle homogenate model increased with storage time in groups with different Na^+^ concentrations. High concentrations of Na^+^ inhibited sarcoplasmic protein phosphorylation. The PKA activity at 24 h of storage and the sarcoplasmic protein phosphorylation level at 12 h of storage in the group with 200 mM Na^+^ was lower than that of the other groups. After 1 h incubation, the PKA activity of samples in the 200 mM Na^+^ group was inhibited and lower than that in the other Na^+^ groups in the in vitro incubation model. These results suggest that the Na^+^ concentration at 200 mM could better inhibit PKA activity. This study provided valuable insights for enhancing curing efficiency and improving meat quality.

## 1. Introduction

cAMP-dependent protein kinase (PKA) is a significant member of the protein kinase family, responsible for catalyzing protein phosphorylation. It consists of two regulatory subunits and two catalytic subunits, forming a tetramer. The catalytic subunit domain can bind and hydrolyze the active site of ATP and interact with the regulatory subunit. The regulatory subunit domain can bind cAMP, the catalytic subunit domain and the self-inhibition region. When the catalytic subunit of PKA dissociates, it phosphorylates specific serine and threonine residues of proteins in the presence of ATP substrates, thereby regulating various cellular functions [1,2]. Previous studies have demonstrated that PKA facilitates the phosphorylation of animal myofibrillar proteins and enhances their degradation [3]. Furthermore, PKA catalysis influences the phosphorylation/dephosphorylation reaction of μ-calpain, resulting in an increase in μ-calpain activity with higher phosphorylation levels [4]. Thus, PKA is significant in the protein phosphorylation process.

Curing has historically been a frequently employed method of processing to prolong the shelf life of meat products. The key component utilized in curing is salt, typically sodium chloride, which not only acts as a preservative but also imparts flavor [5]. Sodium ions (Na^+^) exert influence over enzyme activity in organisms, including the intricate process of protein phosphorylation [6]. In the realm of plants, Na^+^ bolsters protein phosphorylation and plays a pivotal role in the modulation of photosynthesis [7]. Within animal muscles, Na^+^ modifies the extent of protein phosphorylation, an essential factor in regulating metabolism and biological processes. Notably, in an endothelial cell model, the introduction of 20 mM NaCl was found to significantly diminish the activity of the AMPK enzyme and protract the initiation of glycogen degradation [8]. Sodium salts inhibit the activity of various kinases, including PKA in muscle, leading to a decrease in protein phosphorylation levels [9]. The PKA activity of mutton marinated in NaCl was significantly reduced, resulting in a notable decrease in the phosphorylation level of skeletal proteins as well [10]. The presence of Na^+^ influences PKA activity, consequently affecting protein phosphorylation.

Protein phosphorylation primarily affects proteolysis and postmortem glycolytic processes by regulating enzymatic activity, which ultimately determines the quality of meat [11]. In the marinating process, the presence of Na^+^ triggers protein phosphorylation in muscle, thereby impacting the meat’s quality [12]. Studies have shown that the addition of NaCl during curing modifies the phosphorylation levels of various enzymes, leading to an accelerated glycolytic process and improved meat tenderness [8,13]. Pan et al. (2022) found that the preservation process significantly reduced phosphorylation levels in adipocytes, enhancing the hydrolysis of triglycerides within intramuscular fat cells [14]. This phenomenon was predominantly attributed to the activation of protein kinase activity and protein phosphorylation processes. Additionally, it has been observed that administering an injection of 20 mM NaCl within endothelial cells may attenuate the phosphorylation levels of various kinases [8]. However, the appropriate amount of NaCl required to regulate PKA activity and protein phosphorylation during postmortem meat curing is still unclear.

Therefore, the purpose of this study was to investigate the effects of different Na^+^ concentrations on PKA activity and protein phosphorylation levels in postmortem meat. The aim was to reveal the mechanism by which Na^+^ inhibits PKA activity, providing a reference for improving curing efficiency, reducing salt consumption and enhancing meat quality.

## 2. Materials and Methods

### 2.1. Experimental Design

This study consisted of two experiments. The first experiment was carried out to investigate the effects of adding different concentrations of Na^+^ on PKA activity in a muscle homogenate model. The second experiment was to verify that Na^+^ affects PKA activity through an incubation model in vitro. The experimental design was as follows:

Experiment 1: Ten male small tail Han sheep carcasses (7 months old) were randomly chosen from a commercial slaughter plant (Hebei Jinhong Meat Co., Ltd., Hebei, China). The longissimus dorsi muscles were collected within 1 h after slaughter and then stored at −80 °C until analysis. The meat samples of 0.1 g were homogenized twice with 2.1 mL NaCl of 0, 20, 100, 200 and 400 mM, respectively. These homogenized samples were stored at 4 °C. The samples were collected at 0, 1, 3, 12 and 24 h and stored at −80 °C.Experiment 2: An incubation model was established in vitro to confirm the effect of Na^+^ on PKA activity. The incubation model consisted of pure PKA, ATP lysis solution, incubation buffer (10 mM MgCl_2_, 10 mM DTT, 50 mM Tris, pH 6.8) and different concentrations of NaCl (0, 20, 100, 200, 400 mM) added at 4 °C. The model was incubated in a constant temperature metal bath (Ruicheng Instrument Co., Ltd, Hangzhou, China). Samples were collected at 0, 1, 3, 6 and 12 h during incubation.

### 2.2. Activity of PKA

The samples of 0.1 g were homogenized twice with 0.9 mL of 0.1 mol/L phosphate buffer solution (PBS) on ice, then centrifuged at 10,000× *g* (4 °C) for 30 min. The supernatant was collected as the sample to be detected. The activity of PKA was detected using commercial kits (CLP0229, Solarbio, Beijing, China). The reaction system was established according to the manufacturer’s instructions for the PKA activity assay kit. After thorough mixing, absorbance values were measured at 450 nm and the PKA activity was calculated.

### 2.3. Content of cAMP

The samples of 0.1 g were homogenized twice with 0.8 mL of 0.1 mol/L PBS on ice, then centrifuged at 3000× *g* (4 °C) for 30 min. The supernatant was collected as the sample to be detected. The content of cAMP was detected using commercial kits (SEKSM-0017, BC0300, Solarbio, Beijing, China). The reaction system was established according to the manufacturer’s instructions for the cAMP content assay kit. After thorough mixing, the absorbance value was determined at 450 nm and the cAMP content was calculated from the established standard curve.

### 2.4. Content of ATP

The content of ATP was detected using commercial kits (SEKSM-0017, BC0300, Solarbio, Beijing, China). The samples of 0.1 g were homogenized once with 0.9 mL of extract liquid on ice. The homogenized samples were subjected to a water bath at 100 °C for 2 min. After cooling, samples were centrifuged at 10,000× *g* (4 °C) for 10 min. The supernatant was collected as the sample to be detected. The reaction system was established according to the manufacturer’s instructions for the ATP content assay kit. After thorough mixing, the absorbance value was determined by fluorometry and the ATP content was calculated.

### 2.5. Extraction of Sarcoplasmic and Myofibrillar Proteins

The protein extraction method used in this study was based on the procedure described by Lametsch et al. (2006) with slight modifications [15]. Frozen meat samples (1 g) were taken from −80 °C storage and mixed with a buffer solution consisting of 10 mM Tris-HCl, 10 mM DTT, 1 tablet of protease inhibitor and 5 tablets of phosphatase inhibitor. The mixture was then homogenized twice on ice for 10 s each, with a 30 s interval between each homogenization step. After homogenization, the mixture was centrifuged at 10,000 r/min (4 °C) for 30 min. The resulting supernatant was collected as the sarcoplasmic protein sample, while the precipitate was collected as the myofibrillar protein sample. The sarcoplasmic and myofibrillar protein samples were further diluted tenfold, and their protein concentrations were determined using the bicinchoninic acid (BCA) assay. The protein concentration was adjusted to 4 µg/mL. Subsequently, the protein samples were mixed with loading buffer (consisting of 10% SDS, 50% glycerol, 0.5 M Tris-HCl, pH adjusted to 6.8, 1M DTT and 0.01% bromophenol blue) in equal proportions. The samples were then subjected to a water bath at 100 °C for 5 min, followed by collection and storage at −80 °C.

### 2.6. Phosphorylation Levels of Sarcoplasmic and Myofibrillar Proteins

The phosphorylation levels of sarcoplasmic and myofibrillar proteins were measured following the method of Chen et al. (2016), with slight modifications [16]. Electrophoresis was conducted using SDS-PAGE with a 10% separating gel and a 4% concentrating gel. The protein loading volume was 5 μg. Initially, electrophoresis was carried out at 80 V, and the voltage was adjusted to 120 V once the proteins entered the separation gel. Electrophoresis was stopped when the proteins reached 5 mm from the bottom of the gel. All procedures from the end of electrophoresis until gel imaging were performed on a shaker. The gels were fixed twice (30 min each time) in fixative solution (10% acetic acid, 50% ethanol) and eluted three times (10 min each time) with ultrapure water. Subsequently, the cells were stained with Pro-Q Diamond (Invitrogen, Carlsbad, CA, USA) in the dark for 80 min. Afterward, it was decolorized twice (30 min each time) in the dark using Pro-Q decolorization solution (20% acetonitrile, 50 mM sodium acetate, pH 4.0). Phosphoprotein gel imaging was performed after elution with ultrapure water three times (10 min each time). Following this, the gel was stained with SYPRO Ruby (Invitrogen, Carlsbad, CA, USA) in the dark for 4 h. It was then decolorized twice (30 min each time) using SYPRO Ruby decolorization solution (7% acetic acid, 10% ethanol). After washing with ultrapure water (three times, 10 min each time), gel imaging was performed. 

The ChemiDoc^TM^ MP gel imaging system (Bio-Rad, Hercules, CA, USA) was utilized for scanning the images. After staining with Pro-Q Diamond, the images were scanned at an excitation wavelength of 532 nm, an emission wavelength of 580 nm and a resolution of 200 mM. Similarly, after staining with SYPRO Ruby, the images were scanned at an excitation wavelength of 532 nm, an emission wavelength of 610 nm and a resolution of 200 mM. The optical density values of protein bands in SDS-PAGE gel electrophoresis were analyzed using Quantity One software (version 4.62, Bio-Rad, Hercules, CA, USA). For each band, the optical density value (P) of phosphorylated protein staining (Pro-Q Diamond) and the optical density value (T) of whole protein staining (SYPRO Ruby) were obtained. The protein phosphorylation level of the band was determined by calculating the ratio P/T of the optical density values. The overall phosphorylation level of the sample was calculated by summing the P/T values of all protein bands in the lane.

### 2.7. Statistical Analysis

The results were expressed as the mean ± standard deviation (SD) and analyzed by SPSS Statistics 21.0 software (IBM Corp., Armonk, NY, USA). All the experimental data were analyzed using two-way ANOVA. Differences at *p* < 0.05 were considered statistically significant. The pictures of results were captured by Origin 2021 (OriginLab Co., Northampton, MA, USA).

## 3. Results

### 3.1. Changes of PKA Activity at Different Na^+^ Concentrations

The changes of PKA activity in the muscle homogenate model with different Na^+^ concentrations at different storage times are shown in Figure 1 and Appendix A. The PKA activity of samples from the 0, 20 and 100 mM Na^+^ groups was significantly lower than that in the 200 and 400 mM Na^+^ groups at the beginning of storage. The PKA activity of the 100 mM Na^+^ group was higher than that of the 0, 20, 200 and 400 mM Na^+^ groups at storage time of 1 h. The PKA activity of 20, 100 and 200 mM Na^+^ groups was significantly higher than that of 0 and 400 mM Na^+^ groups at storage time of 3 h (*p* < 0.05). In the 200 mM Na^+^ group, the PKA activity at 0 and 1 h was significantly lower than that at 3, 12 and 24 h (*p* < 0.05). The results showed that the PKA activity of the sample in the 200 mM Na^+^ group was reduced at 24 h of storage (*p* < 0.05), indicating that the PKA activity was inhibited when 200 mM Na^+^ was added. The incubation model of pure PKA in vitro is shown in Figure 2 and Appendix A. Na^+^ had different effects on PKA activity depending on the incubation time. After 1 h of incubation, the PKA activity of the treatment group with 200 mM Na^+^ was significantly lower than that of the other treatment groups (*p* < 0.05). This is consistent with the results in the muscle homogenate model. 

### 3.2. Changes in ATP Content in Different Na^+^ Concentrations

The changes in ATP content in the muscle homogenate model with different Na^+^ concentrations at different storage times are shown in Table 1. The ATP content in the 200 and 400 mM Na^+^ group was significantly higher than that in the other groups at the beginning of incubation (*p* < 0.05). And the ATP content in the 0 and 100 mM Na^+^ groups were significantly lower than that in other groups (*p* < 0.05). The ATP content in the 200 mM Na^+^ group was higher than that at 0 and 1 h. 

At the incubation model of pure PKA in vitro, the ATP content of different groups decreased with the extension of incubation time. In the 200 mM Na^+^ group, the ATP content was the highest at 0 h of incubation. And the ATP content at 0–6 h of incubation was significantly higher than that at 12 h of incubation (*p* < 0.05) (Figure 3).

### 3.3. Changes in cAMP Content in Different Na^+^ Concentrations

The changes in cAMP content in the muscle homogenate model with different Na^+^ concentrations at different storage times are shown in Figure 4 and Appendix A. With the extension of storage time, the group without adding Na^+^ gradually decreased and the other groups gradually increased. The cAMP content of 200 mM Na^+^ was lower than that of the other groups at each storage time.

### 3.4. Analysis of Phosphorylation Levels of Sarcoplasmic Protein at Different Na^+^ Concentrations

The changes of sarcoplasmic protein phosphorylation in the muscle homogenate model with different Na^+^ concentrations at different storage times are shown in Figure 5 and Appendix A. The phosphorylation of sarcoplasmic proteins was affected differently by different Na^+^ concentrations at different storage times. The phosphorylation level of sarcoplasmic protein in the 200 mM Na^+^ group was lower than that in the 0, 20, 100 and 400 mM Na^+^ groups after 12 h of storage. Throughout the 0–24 h storage period, the phosphorylation level of sarcoplasmic protein in the 200 mM Na^+^ group after 12 h of storage was lower than at other time points.

### 3.5. Analysis of Phosphorylation Levels of Myofibrillar Protein at Different Na^+^ Concentrations

The changes of myofibrillar protein phosphorylation in the muscle homogenate model with different Na^+^ concentrations at different storage times are shown in Figure 6 and Appendix A. There was no significant difference in myofibrillar protein phosphorylation level at different Na^+^ concentrations at different storage times, and there was no significant difference in myofibrillar protein phosphorylation level at the same Na^+^ at different storage times (*p* > 0.05).

## 4. Discussion

Curing is a traditional method used for food preservation, particularly in preserving meat products and enhancing their quality in terms of water retention and tenderness [17]. Salt not only provides saltiness and enhances flavor during meat curing; it also reduces water activity in meat. Thus, the growth of harmful microorganisms is inhibited, the quality of meat is improved and the preservation time of meat products is prolonged [18]. Salt has an impact on enzyme activity and protein phosphorylation in organisms. After pickling, the phosphorylation level of glycogen phosphorylase increases, facilitating the conversion of the enzyme from inactive to active form. And this process also promotes glycolysis, maintaining the tenderizing effect on meat [19]. PKA, a type of threonine and serine kinase, plays a role in phosphorylating proteins with specific serine and threonine residues in the presence of ATP substrate. Na^+^ is a crucial factor influencing PKA activity. Na^+^ reduces the overall phosphorylation level of meat proteins by inhibiting PKA activity and altering the metabolic process [17]. Furthermore, studies have shown that Na^+^ intake significantly inhibits the activity of PKA and other kinases in muscles, leading to reduced protein phosphorylation levels [9]. Marinating mutton with 2% and 3% NaCl has been found to significantly decrease PKA activity, as well as the phosphorylation levels of muscle glycolytic rate-limiting enzymes and skeleton proteins [17]. The results of this study demonstrate that PKA activity is inhibited when Na^+^ concentration reaches 200 mM in the muscle homogenate model, and the results from the incubation model in vitro confirmed this. 

cAMP is a crucial second messenger in cells, with its intracellular concentration influencing various signal transduction pathways. This, in turn, regulates protein activity, gene expression and other cellular processes such as metabolism, growth and apoptosis [20]. The regulatory effect of cAMP on animal physiological metabolism primarily occurs through enzyme regulation, which involves two distinct pathways. In higher animal cells, cAMP can modulate enzyme activity levels by activating ‘protein kinases’ that catalyze enzyme phosphorylation. This regulates enzyme activity levels. Additionally, cAMP can regulate enzymes by enhancing the transcription process of intracellular mRNA, thereby controlling the expression of the animal’s genetic code. By modulating the production of different enzymes, cAMP achieves its goal of enzyme regulation. Therefore, in animal metabolic activities, cAMP not only regulates enzyme activity but also exerts a certain level of control over the expression of animal genes [21]. The activity of PKA is primarily influenced by the intracellular cAMP concentration [22]. Adenylyl cyclase, which binds to G protein-coupled receptors, is responsible for its activation and catalyzes the synthesis of cAMP from ATP in the cytoplasmic membrane. Once cAMP binds to the regulatory subunit of PKA, it promotes the release of its catalytic subunit. The catalytic subunit then acts on the cAMP response binding protein (CREB), leading to its phosphorylation into the CREB. The CREB mediates neuronal responses to various neurotrophic factors, including brain-derived neurotrophic factor and nerve growth factor [23,24]. In this study, when Na^+^ concentration was 200 mM, PKA activity was inhibited, resulting in a lower cAMP content compared to other Na^+^ concentration groups. During the protein phosphorylation reaction, PKA was activated by consuming the generated cAMP, which subsequently triggered the protein to undergo phosphorylation. Therefore, PKA activity was closely associated with the cAMP content.

Previous studies have demonstrated that PKA activity varies under different Na^+^ concentrations, and a high-salt diet inhibits skeletal muscle cAMP production [25]. And adenylyl cyclase is activated by binding to G protein-coupled receptors in response to various extracellular stimuli and facilitates the synthesis of cAMP from ATP in the plasma membrane. The combination of cAMP with the regulatory subunit of PKA promotes the release of its catalytic subunit, which then catalyzes the phosphorylation reaction [8]. Hence, in the presence of ATP, cAMP can bind to the regulatory subunit of PKA, inducing a conformational change, and releasing the catalytically active catalytic subunit through the entire enzyme, thereby activating PKA and influencing its activity. Furthermore, ATP serves as the energy currency within cells and is continuously hydrolyzed to support cellular functions such as ion channel regulation, signal transduction and gene transcription [26]. In this study, various Na^+^ groups were found to have different ATP contents and PKA activities. When PKA was activated by cAMP, phosphorylation of specific serine/threonine residues in proteins occurred in the presence of ATP, thereby regulating cellular metabolism and gene expression. This suggests that the presence of ATP can influence PKA activity, which in turn affects the protein phosphorylation reaction.

Protein phosphorylation is influenced by a variety of factors, among which protein kinases play a crucial regulatory role. Furthermore, protein phosphorylation leads to the degradation of myofibrillar proteins. In an incubation test conducted by Toyooka et al. (1982), it was discovered that PKA catalysis increases calpain activity, thereby promoting protein phosphorylation and impacting the degradation of myofibrillar proteins [27]. Additionally, Zhang et al. (1988) observed that the presence of PKA changed the phosphorylation of actin-binding proteins [28]. In the present study, the phosphorylation levels of both sarcoplasmic and myofibrillar proteins were altered by the addition of different concentrations of Na^+^. This is consistent with the results of previous studies.

Protein phosphorylation plays an important role in regulating the physiological and biochemical processes of post-mortem muscles, such as glycolysis, muscle contraction, apoptosis, etc., which in turn affects the tenderness, water holding capacity and color stability of meat [16,29,30,31]. Myofibrillar proteins are structural proteins that regulate the structural integrity of sarcomere and muscle contraction. Both the integrity of sarcomere structure and muscle contraction affect the tenderness of meat. Studies have found that salting improves the tenderness of meat [32]. In this study, the phosphorylation level of myofibrillar protein was decreased in the 200 mM Na^+^ group after 1 h of storage. Chen et al. (2016) compared the changes in the phosphorylation level of myofibrillar protein in different tenderness groups. The results showed that the overall phosphorylation level of myofibril protein in the high-tenderness mutton group was lower than that in the low-tenderness group, and protein phosphorylation regulated meat tenderness [16]. This is consistent with the results of this experiment. Many studies have found that phosphorylated myofibrillar proteins are not easily degraded by calpain. After the phosphorylation of troponin by PKA, the sensitivity of the protein to calpain degradation is reduced. The PKA-mediated phosphorylation of troponin is detrimental to troponin degradation [27,33]. Some studies have shown that the increase in Na^+^ affects the phosphorylation level of Ser1054 residues and inhibits the interaction between acyl hydrolase and activated proteins [34], indicating that Na^+^ can affect protein phosphorylation. Protein phosphorylation may be a pathway through which salt curing regulates meat quality. In this study, it was observed that 200 mM NaCl significantly reduced PKA activity, leading to a decrease in sarcoplasmic protein phosphorylation levels. Zhang et al. (2016) conducted a study using myofibrillar protein as a substrate [17]. They established an in vitro model by adding protein kinase A, alkaline phosphatase and salt to different treatment groups. The researchers found that salt had no effect on protein phosphorylation when PKA was not added. This suggests that salt ions and phosphate groups do not competitively bind to protein amino acid residues. Instead, the regulation of protein phosphorylation levels by salt is mainly achieved by inhibiting the activity of enzymes involved in protein phosphorylation reactions. Therefore, the level of PKA activity during the curing process is crucial as it can impact the level of protein phosphorylation. The level of PKA activity is very important during the curing process, which can affect the level of protein phosphorylation and thus the curing effect. Therefore, regulating the Na^+^ concentration during curing could be a promising approach to enhance curing efficiency, reduce salt consumption and improve the overall quality of meat products.

## 5. Conclusions

The activity of PKA was influenced by the presence of Na^+^ concentration in this study. Specifically, PKA activity was inhibited when the Na^+^ concentrations reached 200 mM. The inhibition of PKA activity resulted in a decrease in ATP consumption, which reduced the phosphorylation level of sarcoplasmic proteins consequently. This study provides a reference for improving curing efficiency, reducing salt consumption and improving meat quality.

## Figures and Tables

**Figure 1 foods-13-01647-f001:**
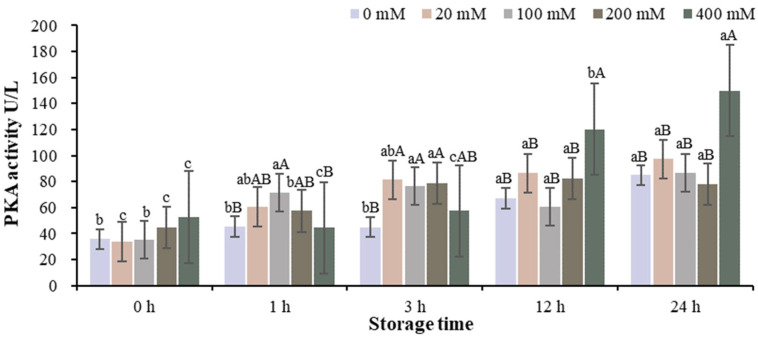
Changes in PKA activity in the muscle homogenate model with different Na^+^ concentrations at different storage times. Note: Different capital letters indicate significant differences at 0.05 level between different groups at the same storage time. Different lowercase letters indicate significant difference at 0.05 level between different times in the same group.

**Figure 2 foods-13-01647-f002:**
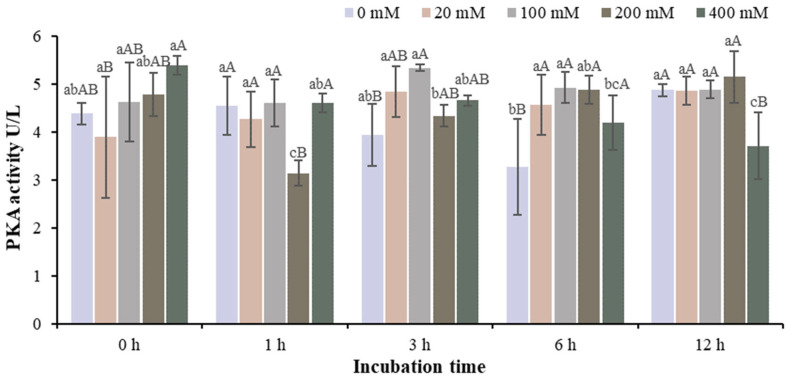
Changes in PKA activity in the incubation model with different Na^+^ concentrations at different incubation times. Note: Different capital letters indicate significant differences at 0.05 level between different groups at the same storage time. Different lowercase letters indicate significant difference at 0.05 level between different times in the same group.

**Figure 3 foods-13-01647-f003:**
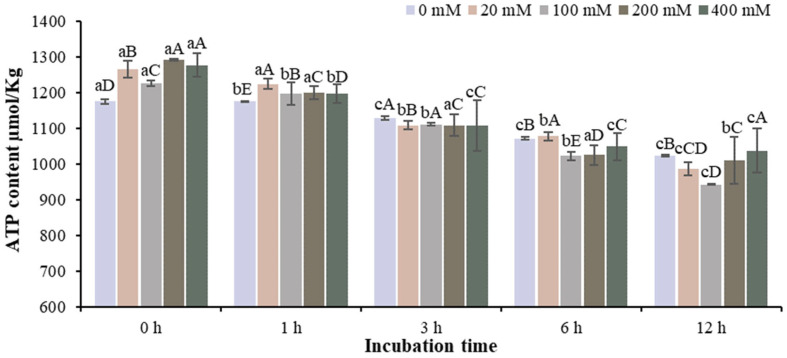
Changes in ATP content in the incubation model with different Na^+^ concentrations at different incubation times. Note: Different capital letters indicate significant differences at 0.05 level between different groups at the same storage time. Different lowercase letters indicate significant difference at 0.05 level between different times in the same group.

**Figure 4 foods-13-01647-f004:**
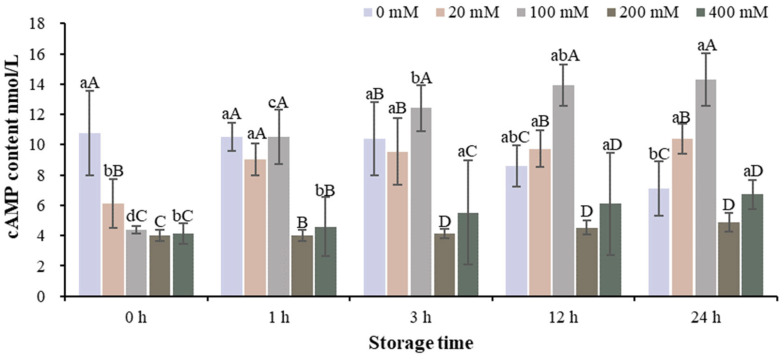
Changes in cAMP content in the muscle homogenate model with different Na^+^ concentrations at different storage times. Note: Different capital letters indicate significant differences at 0.05 level between different groups at the same storage time. Different lowercase letters indicate significant difference at 0.05 level between different times in the same group.

**Figure 5 foods-13-01647-f005:**
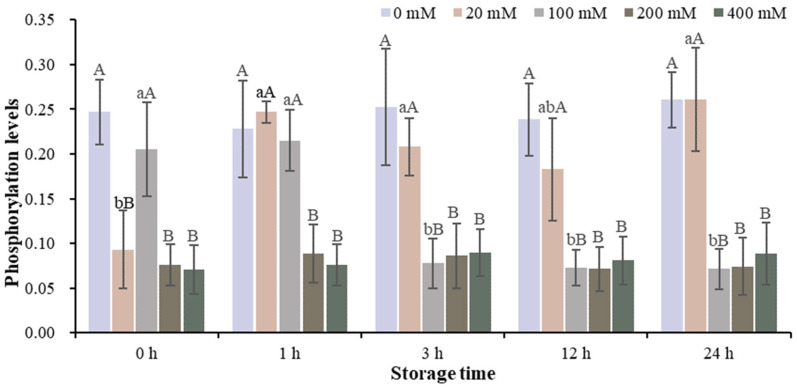
Changes of sarcoplasmic protein phosphorylation in the muscle homogenate model with different Na^+^ concentrations at different storage times. Note: Different capital letters indicate significant differences at 0.05 level between different groups at the same storage time. Different lowercase letters indicate significant difference at 0.05 level between different times in the same group.

**Figure 6 foods-13-01647-f006:**
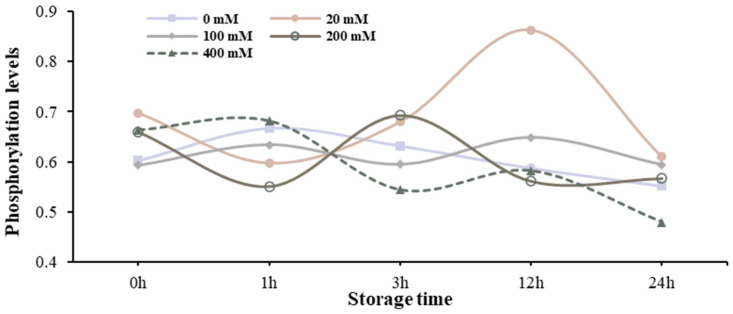
Changes of myofibrillar protein phosphorylation in the muscle homogenate model with different Na^+^ concentrations at different storage times.

**Table 1 foods-13-01647-t001:** Changes in ATP content in the muscle homogenate model with different Na^+^ concentrations at different storage times.

Treatment	0 h	1 h	3 h	12 h	24 h
0 mM	2.54 ± 0.16 ^C^	2.48 ± 0.23	2.47 ± 0.23	2.45 ± 0.04	2.44 ± 0.12
20 mM	2.74 ± 0.15 ^B^	2.69 ± 0.14	2.68 ± 0.14	2.54 ± 0.15	2.51 ± 0.14
100 mM	2.52 ± 0.24 ^C^	2.45 ± 0.25	2.44 ± 0.11	2.60 ± 0.07	2.60 ± 0.14
200 mM	2.81 ± 0.24 ^aA^	2.57 ± 0.12 ^a^	2.52 ± 0.06 ^b^	2.56 ± 0.24 ^a^	2.56 ± 0.10 ^b^
400 mM	2.83 ± 0.17 ^A^	2.64 ± 0.09	2.63 ± 0.16	2.48 ± 0.26	2.40 ± 0.43

Note: Different capital letters indicate significant differences at 0.05 level between different groups at the same storage time. Different lowercase letters indicate significant difference at 0.05 level between different times in the same group.

## Data Availability

The original contributions presented in the study are included in the article/Appendix A, further inquiries can be directed to the corresponding author/s.

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
