# Peer review of "Effects of Different Na+ Concentrations on cAMP-Dependent Protein Kinase Activity in Postmortem Meat"

_foods, 2024, doi:10.3390/foods13111647_

Round 1

Reviewer 1 Report

Comments and Suggestions for Authors

The manuscript shows solidity in its approach and its results are interesting for the scientific community. In the attached document, doubts and details about the investigation are shared.

Author Response

Reviewer #1

Abstract:

Line 9-10: postmortem meat… Is this term correct?

Response: Thank you for your comment! Based on the opinions of the two reviewers, we have corrected the sentence to ensure its correctness. Please see lines 9-10.

Line 12: postmortem meat or postmortem muscle?

Response: Thank you. We have changed “meat” to “muscle”. Please see line 12.

Introduction:

Good argumentation

Response: Thank you for your kind comments!

Materials and Methods:

Justify why muscle was used 1 hour postmortem and not after the maturation process.

Response: Thank you for your comment. We observed the metabolic changes of PKA activity, ATP content and protein phosphorylation level through the physiological state of muscle rigidity to provide theoretical basis for the regulation of meat quality.

0, 20, 100, 200 and 400 mM NaCl are not equally spaced treatments... justify why the order of these treatments.

Response: Thank you for your comment! Through pre-experiments, we found that the effect of adding Na+ concentrations of 0, 20, 100, 200 and 400 mM was better.

Describe the incubation procedure (equipment, model, etc.)

Response: Thank you for your comment. The incubation model consisted of pure PKA, ATP lysis solution, incubation buffer (10 mM MgCl2, 10 mM DTT, 50 mM Tris, pH 6.8), and different concentrations of NaCl (0, 20, 100, 200, 400 mM) added at 4 °C. The model was incubated in a constant tempera-ture metal bath (Ruicheng Instrument Co.,Ltd, Hangzhou). Please see lines 86-89.

Results:

What implications does the reduction of PKA activity have in the first hour of incubation?

Response: Thank you for your comment. In Figure 2, we observed that PKA activity was inhibited when sodium ion was incubated at 200mM for 1 hour, which was consistent with the results of PKA activity inhibition in muscle homogenate model, demonstrating the existence of this physiological state in postmortem muscle.

What is the significance of the reduction in sarcoplasmic protein phosphorylation in 200 mM Na+ during the incubation period?

Response: Thank you for your comment. We re-analyzed the data, throughout the 0-24 h storage period, the phosphorylation level of sarcoplasmic protein in the 200 mM Na+ group after 12 hours of storage was lower than at other time points. Please see lines 238-240.

Discussion:

Good argumentation

Response: Thank you for your kind comments!

Conclusions:

The conclusions are concrete, however, it is worth commenting on the practical implications in the curing and/or marinating of meat through the use of sodium in the concentration favorably described in the present study.

Response: Thank you for your comment. This study provides a reference for improving curing efficiency, reducing salt consumption and improving meat quality. Please see lines 365-367.

Supplementary material

Table S1. The quality indicator results of muscle homogenate model/incubation model in different Na+ concentrations at different time.

Index

P-value

Time

Na+

Time ×Na+

PKA activity

(muscle homogenate model)

<0.001

<0.001

<0.001

PKA activity

(incubation model)

0.107

0.027

<0.001

ATP content

(incubation model)

<0.001

0.004

<0.001

cAMP content

<0.001

<0.001

<0.001

phosphorylation levels of sarcoplasmic protein

0.053

<0.001

<0.001

phosphorylation levels of myofibrillar protein

0.738

0.653

0.986

Reviewer 2 Report

Comments and Suggestions for Authors

There are only a few language errors. The manuscript does not make clear in the conclusions how the level of sodium and thus PKA activity can be adjusted or controlled to more optimize meat quality properties in cured or salted meat.

 Line(s)      Comment

9-10          “cAMP-dependent protein kinase (PKA) activity regulates protein phosphorylation, with Na+ playing a crucial role in PKA activity. The aim”

15-16        “activity. The PKA activity of the muscle’

16-17        Use of “significantly” and the probability level in the same sentence here and in subsequent sentences is redundant.

83-84        It must be clarified if the homogenization with 2.1 mL NaCl was with the frozen samples or thawed samples. If with thawed samples, then the thawing time and temperature must be given.

89              “400 mM) added at 4°C.”

93              “ice, then centrifuged”

100-101   “ice, then centrifuged”

125           Acronyms and abbreviations should be defined at first use.

165-166   One-way ANOVA is not appropriate because the ANOVA should include replication, Na+ concentrations, storage times, and their interactions. If the replication and the interaction of replication x Na+ level and replication x storage time are not significant, this should be stated in the manuscript and a two-way ANOVA (Na+, storage times, and Na+ x storage time) should be conducted for reporting of the results.

187           Figure 1 clearly shows that there might have been interactions between storage time and the Na+ concentrations, so an ANOVA that determines the level of probability for the interactions must be conducted.

194           Figure 2 clearly shows that there might have been interactions between storage time and the Na+ concentrations, so an ANOVA that determines the level of probability for the interactions must be conducted.

196           The same note on significance differences with different letters in Figure 1 is needed for Figure 2.

209           The same note on significance differences with different letters in Figure 1 is needed for Table 1.

212           The same note on significance differences with different letters in Figure 1 is needed for Figure 3.

221           The same note on significance differences with different letters in Figure 1 is needed for Figure 4.

225-226   This sentence is indicative of the need to conduct a 2 factor ANOVA.

233           The same note on significance differences with different letters in Figure 1 is needed for Figure 5.

241           Letters to indicate differences among means in Figure 6 similar to previous figures are needed.

258           Delete “can”

297-299   Incomplete sentence.

352-355   The conclusion only reiterates the results and does not give conclusions as to the use of the results, why they are innovative or useful, and/or additional research needed. It was expected that additional conclusions based upon l. 347-350 would be provided, including recommendations on the appropriate levels of salt in curing solutions and/or NaCl amounts added during the curing process, and to what extend NaCl concentrations could be decreased and still maintain the desired levels of water holding, microbial inhibition, and tenderization.

373, 413, 444 The scientific journal names are not abbreviated in the other references.

386, 439, 452 The journal name is not in all capital letters in other references.

Comments on the Quality of English Language

There are only a few language errors.

Author Response

Reviewer #2

9-10 “cAMP-dependent protein kinase (PKA) activity regulates protein phosphorylation, with Na+ playing a crucial role in PKA activity. The aim”

Response: Thank you. We have made changes. Please see lines 9-10.

15-16 “activity. The PKA activity of the muscle’

Response: Thank you. We have made changes. Please see line 15.

16-17 Use of “significantly” and the probability level in the same sentence here and in subsequent sentences is redundant.

Response: Thank you. We have made changes. Please see line 15-16.

83-84 It must be clarified if the homogenization with 2.1 mL NaCl was with the frozen samples or thawed samples. If with thawed samples, then the thawing time and temperature must be given.

Response: Thanks for your comment! Meat samples are stored at -80 °C. It is a frozen sample and does not undergo thawing. Before the experiment, 0.1g was quickly cut, NaCl was added and homogenized at 4 ℃.

89 “400 mM) added at 4°C.”

Response: Thank you. We have made changes. Please see line 89.

93 “ice, then centrifuged”

Response: Thank you. We have made changes. Please see line 94.

100-101 “ice, then centrifuged”

Response: Thank you. We have made changes. Please see lines 101-102.

125 Acronyms and abbreviations should be defined at first use.

Response: Thanks for your comment! We have made changes. Please see lines 125-126.

165-166 One-way ANOVA is not appropriate because the ANOVA should include replication, Na+ concentrations, storage times, and their interactions. If the replication and the interaction of replication x Na+ level and replication x storage time are not significant, this should be stated in the manuscript and a two-way ANOVA (Na+, storage times, and Na+ x storage time) should be conducted for reporting of the results.

187 Figure 1 clearly shows that there might have been interactions between storage time and the Na+ concentrations, so an ANOVA that determines the level of probability for the interactions must be conducted.

194 Figure 2 clearly shows that there might have been interactions between storage time and the Na+ concentrations, so an ANOVA that determines the level of probability for the interactions must be conducted.

225-226 This sentence is indicative of the need to conduct a 2 factor ANOVA.

Response: Thank you for your comment. We have supplemented the two-way ANOVA results according to your requirements, details are shown in the supplementary material (table S1). The relevant descriptions of the results and pictures have been corrected in the revised draft.

196 The same note on significance differences with different letters in Figure 1 is needed for Figure 2.

209 The same note on significance differences with different letters in Figure 1 is needed for Table 1.

212 The same note on significance differences with different letters in Figure 1 is needed for Figure 3.

221 The same note on significance differences with different letters in Figure 1 is needed for Figure 4.

233 The same note on significance differences with different letters in Figure 1 is needed for Figure 5.

Response: Thank you. We have made changes.  

241 Letters to indicate differences among means in Figure 6 similar to previous figures are needed.

Response: Thank you for your comment. According to the results of two-way ANOVA analysis, Time×Na+ had no significant difference in the phosphorylation level of myofibrillar protein. By one-way ANOVA analysis, there was only one result with a significant difference. Therefore, we pay more attention to changes in protein phosphorylation levels.

258 Delete “can”

Response: Thank you. We have made changes. Please see line 269.

297-299 Incomplete sentence.

Response: Thank you. We have made changes. Please see lines 307-309.

352-355 The conclusion only reiterates the results and does not give conclusions as to the use of the results, why they are innovative or useful, and/or additional research needed. It was expected that additional conclusions based upon l. 347-350 would be provided, including recommendations on the appropriate levels of salt in curing solutions and/or NaCl amounts added during the curing process, and to what extend NaCl concentrations could be decreased and still maintain the desired levels of water holding, microbial inhibition, and tenderization.

Response: Thank you for your comment. This study provides a reference for improving curing efficiency, reducing salt consumption and improving meat quality. Please see lines 365-367.

373, 413, 444 The scientific journal names are not abbreviated in the other references.

Response: Thank you. We have made changes. Please see lines 385-386, 426, and 458.

386, 439, 452 The journal name is not in all capital letters in other references.

Response: Thank you. We have made changes. Please see lines 399, 453, and 467.

Supplementary material

Table S1. The quality indicator results of muscle homogenate model/incubation model in different Na+ concentrations at different time.

Index

P-value

Time

Na+

Time ×Na+

PKA activity

(muscle homogenate model)

<0.001

<0.001

<0.001

PKA activity

(incubation model)

0.107

0.027

<0.001

ATP content

(incubation model)

<0.001

0.004

<0.001

cAMP content

<0.001

<0.001

<0.001

phosphorylation levels of sarcoplasmic protein

0.053

<0.001

<0.001

phosphorylation levels of myofibrillar protein

0.738

0.653

0.986

Round 2

Reviewer 2 Report

Comments and Suggestions for Authors

The revisions are appreciated. There are some spelling corrections:

Line(s)      Comment

186           The ordinate (y) axis label should be “PKA activity U/L”

398           Electrophoresis